# Methionine supplementing effects on intestine, liver and uterus morphology, and on positivity and expression of Calbindin-D28k and TRPV6 epithelial calcium carriers in laying quail in thermoneutral conditions and under thermal stress

**Lanuza Ribeiro de Moraes**[1], **Maria Eduarda Araújo Delicato**[2], **André da Silva Cruz**[2], **Hugo Thyares Fonseca Nascimento Pereira da Silva**[2], **Clara Virgínia Batista de Vasconcelos Alves**[2], **Danila Barreiro Campos**[1], **Edilson Paes Saraiva**[3], **Fernando Perazzo da Costa**[3], **Ricardo Romão Guerra**[1,3]*

1 Programa de Pós-Graduação em Ciência Animal, Universidade Federal da Paraíba, Areia, Paraíba, Brazil, 2 Departamento de Ciências Agrárias, Universidade Federal da Paraíba, Areia, Paraíba, Brazil, 3 Programa de Pós-Graduação em Zootecnia, Universidade Federal da Paraíba, Areia, Paraíba, Brazil

* rromaoguerra@gmail.com

## Abstract

This study aimed to provide the performance, localization and expression of the epithelial calcium transporter channels Calbindin-D28k (Calb) and TRPV6, and of the morphology of the digestive and reproductive system of laying quail under heat stress (HS), and with methionine supplementation (MS). This study characterized the positivity (immunohistochemistry) and expression (real-time PCR) of calcium channels in the kidneys, intestine and uterus of 504 laying quails under different MS (100, 110 and 120%) and temperatures (20, 24, 28 and 32˚C). The animals under HS (32˚C) had lower villus height, villus:crypt ratio, and goblet cell index in the duodenum and jejunum, fewer secondary and tertiary uterine folds, smaller hepatic steatosis, and increased number of distal convoluted renal tubules (CT) positive to Calb, and increased positivity in proximal CTs. Deleterious effects of HS were minimized with MS for: duodenal crypts, number of goblet cells of the jejunum, number of uterine folds, decreased Calb positivity in intestines and kidney, increased positivity of Calb in the uterus and increased TRPV6 gene expression in the kidney (P≤0.05). Epithelial calcium transporters were altered due to less need for calcium absorption and reabsorption due to more calcium available with the MS, increasing egg production in HS and quality in termoneutrality (P≤0.05). MS further increased intestinal villus absorption area and height, increased steatosis, decreased Calb positivity in the intestine and kidney, increased uterine positivity of Calb, and increase Calb and TRPV6 expression in the kidney (P≤0.001) under thermoneutrality. It was concluded that the use of MS (120%) is justifiable in order to partially reverse the deleterious effects of HS on the production, in the epithelial calcium carriers, and in the digestory and reproductive morphology of laying quail.

**Data Availability Statement:** All relevant data are within the manuscript. Some additional data requested from the real-time PCR analyzes was sent to Plos One during the first referee assessments.

**Funding:** The researcher Guerra, R. R. had a Research Productivity Grant from the National Council for Scientific and Technological Development (CNPq-Brazil), n. 304137/2015-4 and funds through Public Notice 71/2013 PROCAD of the Coordination for the Improvement of Higher Education Personnel (CAPES-Brazil), n. 881.068412 / 2014-01. The funders had no role in study design, data collection and analysis, decision to publish, or preparation of the manuscript. Only the Coordinator of the research received a grant and funds from federal public agencies that aim to develop research in Brazil.

**Competing interests:** The authors have declared that no competing interests exist.

# Introduction

Quail laying farming has been growing in Brazil, mainly in the northeast region. The raising of quails is a very profitable activity and with broad perspectives, which induces the development of research aiming at better production, perfecting techniques and alternatives to reach quality standards and expansion throughout the territory.

In tropical climates, such as those found in most of Brazil, laying birds suffer a reduction in their zootechnical indexes as well as an increase in mortality as a result of thermal stress by heat, leading to productive and economic losses in production [1]. This fact is mainly observed in laying quail farming, where the first restrictive factor for eggshell formation is calcium, and calcium is negatively influenced by temperature increase [2].

Calcium comes mainly from intestinal absorption and bone and kidney resorption, which is mobilized from the blood to the uterus very quickly [3–5] two importants ephitelium calcium carries are Calbindin-D28k and TRPV6 [4–7]. Nascimento et al. [8] stated that 70% of the production cost is based on food, and for this reason there is a need to develop balanced diets according to the needs of the birds, enabling them to use the diet with maximum efficiency.

For birds subjected to heat stress, it is necessary to supplement the diets with glycogenic amino acids, such as methionine, cystine and others, wich would alleviate the deleterious effects of thermal stress [9]. Under such conditions of thermal stress, physiological and behavioral changes occur in quail, which severely affect feed intake and cause structural changes in the intestinal epithelium, reducing nutrient digestibility and absorption [10].

Methionine, classified as an essential amino acid [11], is also the first limiting factor in poultry feed, and is essential for the maintenance, growth, production, and development of feathers [12]. In addition to the productive responses obtained with methionine supplementation in laying hens, studies of the morphological analyses of the laying digestive and reproductive system of broilers and light birds indicate favorable quantitative and qualitative changes, such as an increase in egg mass by 10% [13, 14], decreased laying fat [15], increased eggshell thickness [16], and increased intestinal villi, which provide and technically justify the improvement in zootechnical indexes of these animals with methionine use [17–19].

Evaluating the existing literature [17–19], diets supplemented with methionine at levels above National Research Council (**NRC**) recommendation, can be a nutritional strategy to minimize heat stress damage, by improving the performance of laying birds in warmer regions. Such studies are scarce in laying hens and even more so in laying quail.

Thus, due to the gap in research related to the subject described, the objective was to evaluate the effect of methionine supplementation on intestinal morphology; on the villus:crypt ratio; on the quantity of goblet cells; on liver glycogen storage and steatosis; on uterine morphology in thermoneutral laying quail, and under high temperature heat stress. Furthermore, the objective was to evaluate positivity and gene expression of Calbindin-D28k and TRPV6 in laying birds under the same conditions, to evaluate the alteration of these channels due to methionine supplementation in thermoneutral laying birds and under thermal stress, since they are described as the main epithelial calcium carriers, acting on laying quail intestines, kidneys and uterus [7, 20].

# Material and methods

A total of 504 Japanese quail in production stage (second cycle; 21 days each cycle) from the eighth week of life were used, distributed in a completely randomized design in a 3x4 factorial scheme, with three levels of methionine (100%, 110% and 120%) and four temperature ranges (20 24, 28 and 32˚C), with 07 replications with 06 birds each (42 animals for treatment), representing thermoneutral (20, 24 and 28˚C) and thermal stress ranges (32˚C), totaling twelve

**Table 1. Experimental diets containing three levels of methionine supplementation for laying quail.**

| Ingredientes, % | T1 | T2 | T3 |
|---|---|---|---|
| | 100% Met+Cys (0.888%) | 100+10%Met+Cys (0.977%) | 100+20% Met+Cys (1.066%) |
| Corn, 788% | 58.466 | 58.466 | 58.466 |
| Soybean meal, 45% | 30.071 | 30.071 | 30.071 |
| Soyabean oil | 0.906 | 0.906 | 0.906 |
| Calcitic limestone | 7.041 | 7.041 | 7.041 |
| Dicalcium phosphate, 18.5% | 1.142 | 1.142 | 1.142 |
| Salt | 0.326 | 0.326 | 0.326 |
| L-Lisin HCl | 0.342 | 0.342 | 0.342 |
| L-Threonine | 0.078 | 0.078 | 0.078 |
| L-Tryptophan | 0.040 | 0.040 | 0.040 |
| Choline chloride, 60% | 0.070 | 0.070 | 0.070 |
| Mineral premix | 0.050 | 0.050 | 0.050 |
| Vitaminic Premix | 0.025 | 0.025 | 0.025 |
| Antioxidant | 0.010 | 0.010 | 0.010 |
| DL-Methionine | 0.441 | 0.537 | 0.633 |
| Starch | 0.673 | 0.536 | 0.400 |
| Inert | 0.319 | 0.360 | 0.400 |
| TOTAL, kg | 100.00 | 100.00 | 100.00 |
| **Chemical composition** | | | |
| PB, % | 18.80 | 18.80 | 19.00 |
| EM, kcal/kg | 2800 | 2800 | 2800 |
| Met digestible, % | 0.685 | 0.779 | 0.870 |
| Met + Cis digestible, % | 0.942 | 1.036 | 1.130 |
| Lis digestible, % | 1.148 | 1.148 | 1.148 |
| Thre digestible, % | 0.701 | 0.701 | 0.701 |
| Val digestible, % | 0.785 | 0.785 | 0.785 |
| Trp digestible, % | 0.241 | 0.241 | 0.241 |
| Arg digestible, % | 1.152 | 1.152 | 1.152 |
| Ile digestible, % | 0.717 | 0.717 | 0.717 |
| Leu digestible, % | 1.485 | 1.485 | 1.485 |
| Calcium, % | 2.99 | 2.99 | 2.99 |
| Available match, % | 0.31 | 0.31 | 0.31 |
| Sodium, % | 0.15 | 0.15 | 0.15 |
| Chlorine, % | 0.24 | 0.24 | 0.24 |
| Potassium, % | 0.72 | 0.72 | 0.72 |
| BE, mEq/kg | 179.00 | 179.00 | 179.00 |

treatments. Diets were formulated according to Table 1. The quail started the treatments at eight weeks of age and remained for two cycles (42 days), totaling 98 days of experiment.

The experiment was conducted in Poultry Sector, Department of Animal Science at Federal University of Paraíba. The project had ethical approval from the Animal Use and Care Committee (CEUA) of the Federal University of Paraíba, Brazil, under protocol number 149/2015, and the euthanasia was performed by electronarcosis according to CEUA guidelines.

## Histological processing

Histological processing was performed at the Histology Laboratory of the Center for Agrarian Sciences of the Federal University of Paraíba. Biological samples of intestine (duodenum and

jejunum), liver, uterus and kidney from 8 randomly chosen animals from each treatment (at least 01 animal of each replication) were collected and fixed in Metacarn (60% methanol, 30% chloroform and 10% acetic acid) for 12h and embedded in paraffin in 98˚ day of experiment. The cuts were made with 5μm thickness. Hematoxylin-eosin staining and Periodic Acid Schiff (PAS) were used depending on the analysis, and digitized images were captured on an Olympus BX-60 microscope and a Olympus camera coupled with a Olympus cellSens Dimension digital imaging program. Samples of the duodenum were collected 4 cm after the ventricle and samples of the jejunum were collected in the middle region of this segment. Both were included in a transverse direction, so that it was possible to visualize the intestinal villi as well as the lumen of the organ. Kidney and liver samples were collected so as never to exceed 0.5 $cm^3$ for adequate tissue fixation. Uterine samples were collected in the middle lateral region with a dimension of 1 $cm^2$.

Histomorphometry analyses were performed by a single histologist to avoid misinterpretation. ANOVA and Tukey's post test at 5% significance level were used to evaluate the influence of methionine supplementation on different temperature types. These analyses were performed by using the SAS[®] University Edition [21].

## Duodenal and jejunal morphology

Five photomicrographs were digitized in eight animals from each treatment, and two measurements in each image, totaling 80 measurements (8 animals x 5 photomicrographs x 2 measurements) of intestinal villi height and their respective crypts from each treatment, using an Olympus cellSens Dimension image analyzer and a Olympus digital camera attached to an Olympus BX-60 microscope. Villus height measurements were taken from its base to its apex; the width was measured at the middle portion of each villus, and the crypt was measured from the base of its respective villus. The villus:crypt ratio (VCR) was given by dividing the villus height by its respective crypt.

To quantify the goblet cell index in the duodenum and jejunum epithelium, histological slides from eight animals per treatment stained with PAS staining magenta on the goblet cells were used. Images were captured and digitized with the 20x objective of the intestinal villi. At least 2 images from each animal were randomly chosen and the intestinal epithelium was measured linearly to 2000 μm, and the number of goblet cells per 500 μm was counted, making a sample of 32 per treatment (8 animals x 4 areas of 500 μm). Based on the results, the number of goblet cells in 1,000 micrometers for each treatment was defined based on a rule of three.

## Measurement of hepatic glycogen storage and hepatic steatosis

For the measurement of hepatic glycogen storage, after the aforementioned histological processing and PAS staining, 5 photomicrographs of 8 animals from each treatment, chosen at random, were analyzed by optical microscopy by the same histologist, without his previous knowledge about the group belonging to each bird, totaling a sample of 40 per treatment (8 animals x 5 photomicrographs). The photomicrographs were classified according to the degree of glycogen deposition due to the positivity to PAS histochemistry: Grade +: little hepatic glycogen deposition; Grade ++: moderate hepatic glycogen deposition; and Grade +++: marked hepatic glycogen deposition. For analysis of the hepatic glycogen deposition index, the crosses were transformed into corresponding numbers (+ = 1, ++ = 2, +++ = 3) to perform the statistics according to the modified Ishak Semi Quantitative Score [22].

To evaluate hepatic steatosis, an evaluation score was assigned to each liver analyzed by liver photomicrographs of each animal (8 animals per treatment), totaling a sample number of 40 per treatment (8 animals x 5 photomicrographs), considering the amount and the size of

the hepatocyte lipid cytoplasmic vacuoles: 0 (absence of steatosis), 1 (low steatosis), 2 (moderate steatosis) and 3 (advanced steatosis), following the modified Ishak Semi Quantitative Score [22]. For each treatment an average was obtained, which was submitted to statistics.

## Uterine morphology

After histological processing and the PAS histochemical staining, digital images were captured. Photomicrographs, 5 of each one of the 8 animals per treatment, totaling a sample number of 40, were evaluated according to the presence and quantity of uterine folds, and were evaluated according to modified Ishak Semi-Quantitative Score [22]. Score 1 was given for the presence of primary folds only, score 2 for primary and some secondary folds, score 3 for the presence of primary, secondary and some tertiary folds, and score 4 for uterus with numerous tertiary folds. The photomicrographs were analyzed under optical microscopy by the same histologist, without his previous knowledge about the group belonging to each bird.

## Immunohistochemistry for anti-Calbindin-D28k

Histological slides containing duodenum, jejunum, uterus and kidney from 6 animals per treatment were chosen randomly. The antibody was Calbindin-D28k (Sigma, Clone Cl3000). The slides were dehydrated, blocked with 3 hydrogen peroxide baths for 10 minutes each and washed with phosphate buffer (PBS) 3 times for 3 minutes. The unmasking was performed with citrate buffer (pH 6.0) for 10 minutes in the microwave, waiting for the temperature to drop for another 20 minutes. The slides were again washed in PBS and incubated at 4°C overnight with antibody diluted in PBS (1:200). The slides that received the anti-Calbindin-D28k antibody were incubated with positive control, and those that received only the PBS with negative control. The following day, the slides were placed biotinylated secondary antibody for 15 minutes, followed by incubation in streptavidin peroxidase complex (DAKO-LSAB) for 30 minutes. Positive cells were labeled by diaminobenzidine (DAB) chromogen (DAKO) for 5 minutes. The photomicrographs were performed by the KS-400 Zeiss programs under Olympus BX60 microscope and AxioCam camera. The more antibody-positive, the higher the protein production of Calbindin-D28k. Therefore, positivity can be seen in the tissue by staining the DAB chromogen by brown, so the brown staining demonstrates the location of the Calbindin-D28k protein in the tissue, which is a cytoplasmic protein.

## Real-time PCR (qPCR) for TRPV 6 and Calbindin-D28k

Duodenum and jejunum, kidney and uterus tissues samples were collected from 6 randomly selected animals per treatment. Samples were immediately frozen in liquid nitrogen and then transferred to a -80°C ultra-freezer until processing. Tissues were homogenized in lysis buffer and RNA extraction was performed using the PureLink™ RNA Mini Kit (Thermo Fisher Scientific) and the ReliaPrep™ RNA Tissue Miniprep System (Promega). Concentration and purity of the RNA samples were determined by the absorbance ratio 260/280 and 260/230 obtained using a spectrophotometer (Colibri Microvolume Spectrometer, Titertek Berthold). RNA integrity was verified on agarose gel. Reverse transcription was performed using the SuperScript™ IV VILO™ Master Mix with ezDNase™ Enzyme (Invitrogen) according to the manufacturers' recommendations.

Expression of selected genes was accessed using a Stratagene Mx3005P qPCR (quantitative polymerase chain reaction) System (Agilent Technologies). Oligonucleotides were obtained from Japanese quail sequences previously published and beta actin was used as endogenous control (Table 2). PCR reactions were performed using Brilliant III Ultra-Fast SYBR® Green qPCR Master Mix with Low ROX (Agilent Technologies). Standard curve for each primer was

**Table 2. The sequence of primers used for quantitative polymerase chain reaction (qPCR)–real time for quail.**

| Genes | qPCR Primers (5'-3') | GenBank Number |
|---|---|---|
| Calbindin28 | >GACGGCAATGGGTACATGGA<br><TCGGGTGTTAAGTCCAAGCC | XM_015855985.1 |
| TRPV6 | >CCATCATTGCCACCCTCCTT<br><AGCAACAATCTGGGCTCTCC | XM_015873874.1 |
| Beta Actina | >**A**CCACTGGCATTGTCATGGACTCT<br><ACCACTGGCATTGTCATGGACTCT | XM_015872688.2 |

> = forward, < = reverse

set up using a pool of all experimental cDNA samples in seven 2x dilution from 25 ng to 0.39 ng. The slope of the standard curves were between 3.08 and 3.3, $R^2$ value was above 0.99 and qPCR efficiency range between 100 and 110.8%. Amplification conditions were 3 minutes at 95˚C followed by 40 cycles of 15 seconds at 95˚C and 20 seconds at 60˚C; 1 minute at 95˚; 30 seconds at 55˚, and finally 30 seconds at 95˚.

Melting curve analysis allowed the evaluation of the primer specificity. Samples were run in triplicate for each sample and relative quantification (target gene/endogenous control) determined their expression. Data were normalized to a calibrator sample using the ΔΔCt method [23] with correction for amplification efficiency.

## Statistical analysis

Data were subjected to two way analysis of variance (ANOVA) and, according to the significance of the F test ($P \leq 0.05$), the means were compared by the SNK test (Student-Newman-Keuls) at up to 5% probability of error. These analyses were performed by using the SAS® University Edition [21]. Pearson correlation analysis was also performed using the JMP® Pro 13. Data were represented by means ± standard deviation.

## Results and discussion

### Histology

**Duodenal and jejunal morphology.** From the histomorphometric analysis of the intestine, it was found that the villus height variable (AV) in laying quail is reduced during heat stress in the duodenum and jejunum (Table 3), corroborating Mitchell and Carlisle [24], who observed decrease in jejunal villus height of broilers kept under constant thermal stress compared to birds in thermoneutrality.

Supplementation with 120% methionine provided in duodenum higher villus height (**VH**) at thermoneutral temperatures of 20 and 24˚C, but not at higher temperatures, including thermal stress temperatures (32˚C). Thus, methionine supplementation is ineffective at reversing the harmful effects of heat stress for VH. Supplementation with 120% methionine even led to decreased VHs at the temperature of 32˚C. However, this characteristic can be interpreted in another way. The decrease in villous height can be interpreted as a result of the greater disponibility of nutrients that methionine supplementation provides, including calcium, thus not requiring a large energy expenditure to provide an increase in villi. In this context, it is emphasized that the animal under thermal stress already has a lower food intake. This energy saving can be reverted to animal production, quantitatively or qualitatively. In the present study, there was an improvement in egg quality (Table 4).

In the jejunum, the effects were similar; supplementation with 120% methionine promoted VH increase at 20˚C, maintained VH at 24˚C, and also reduced VH at 28 and 32˚C.

**Table 3. Morphometry of the digestive and breeding system of laying Japanese quail (*Coturnix japonica*) supplemented (S) with methionine at levels of 100%, 110% and 120% submitted to different temperatures (20˚, 24˚, 28˚ and 32˚C).**

| Variable | S % | Temperature (˚C) | | | |
|---|---|---|---|---|---|
| | | **20** | **24** | **28** | **32** |
| Duodenum VH | 100 | 860.85±140.49aB | 808.93±73.59bB | 811.22±97.36bA | 822.56±107.17abA |
| | 110 | 673.9±106.13cC | 856.47±90.46aA | 829.90±127.27aA | 757.96±73.81bB |
| | 120 | 937.65±113.96aA | 886.30±90.76bA | 760.13±67.55cB | 729.15±68.09dC |
| Jejunum VH | 100 | 631.19±95.47bB | 678.79±60.45aA | 653.87±83.03abA | 676.65±77.91aA |
| | 110 | 628.78±98.38aB | 564.32±51.51bB | 578.69±129.69bB | 575.78±85.11bB |
| | 120 | 691.92±66.93aA | 656.45±69.34bA | 547.94±113.41cB | 557.35±77.37cB |
| Duodenum CD | 100 | 46.37±9.69bB | 54.43±11.33aA | 49.97±9.60abA | 45.25±7.04bcAB |
| | 110 | 49.97±9.94aB | 48.92±8.21aB | 47.77±8.67aAB | 44.17±6.88bB |
| | 120 | 57.40±13.31aA | 48.93±8.91bB | 44.84±7.73cB | 47.57±7.85bcA |
| Jejunum CD | 100 | 37.27±5.89cB | 40.60±7.68bB | 45.79±7.53aA | 44.59±6.70aA |
| | 110 | 39.72±6.91bAB | 39.83±6.47bB | 40.83±5.90abB | 42.36±4.78aA |
| | 120 | 41.12±7.64bA | 44.84±7.33aA | 41.11±8.03bB | 44.13±6.02aA |
| Villous: Crypt Duodenum | 100 | 19.38±5.20aA | 15.49±3.38bB | 16.66±3.08bA | 18.45±2.91abA |
| | 110 | 14.05±3.69bC | 17.93±3.20aA | 17.89±4.01aA | 17.58±3.28aA |
| | 120 | 16.88±2.92bcB | 18.84±3.57aA | 17.41±3.16bA | 15.71±2.85cB |
| Villous: Crypt Jejunum | 100 | 17.16±2.79aA | 17.18±2.93aA | 14.61±2.74bA | 15.52±2.94bA |
| | 110 | 15.99±2.13aB | 14.55±2.83abB | 14.40±3.55abA | 13.78±2.67bB |
| | 120 | 17.35±3.41aA | 14.98±2.68bB | 13.46±2.24bcA | 12.80±2.17cB |
| GC Duodenum | 100 | 27.71±8.56abB | 31.38±7.65aA | 30.50±9.93aA | 23.33±7.61bB |
| | 110 | 42.79±15.33aA | 31.08±6.66bcA | 36.00±6.78bA | 27.21±4.23cAB |
| | 120 | 33.08±9.05aB | 25.13±8.36bB | 35.58±7.92aA | 30.46±6.09abA |
| GC Jejunum | 100 | 36.25±6.53abA | 32.67±7.88bB | 40.13±11.16aA | 43.42±11.99aA |
| | 110 | 39.17±10.56bA | 49.04±12.16aA | 38.42±8.80bA | 39.75±7.89bA |
| | 120 | 39.38±10.73aA | 42.88±11.22aA | 40.21±8.06aA | 44.67±8.84aA |
| Uterine folds | 100 | 3.11±0.68aA | 3.12±0.99abA | 2.87±0.64bcB | 2.28±0.83cA |
| | 110 | 3.06±0.73aA | 3.22±0.73aA | 3.06±1.06aA | 2.39±0.61bA |
| | 120 | 3.43±0.51bA | 3.50±0.71aA | 2.28±0.75bB | 2.83±0.71bA |
| Hepatic steatosis | 100 | 1.17±0.59abA | 1.43±0.97aA | 0.70±0.75bB | 0.87±0.78abA |
| | 110 | 1.07±0.64aA | 0.87±1.07aB | 0.60±0.67aB | 0.90±0.80aA |
| | 120 | 1.10±0.84abA | 1.57±1.19aA | 1.43±1.04abA | 0.87±1.01bA |
| Hepatic glycogen | 100 | 1.17±0.38aA | 1.30±0.53aB | 1.43±0.63aA | 1.43±0.63aA |
| | 110 | 1.17±0.38bA | 2.53±0.63aA | 1.03±0.18bB | 1.27±0.45bA |
| | 120 | 1.43±0.50aA | 1.00±0.00bB | 1.50±0.51aA | 1.43±0.68aA |

Averages followed by the differente letter lowercase in the row and uppercase in the column differ (P<0,05%) probability; VH: villus height; CD: crypt depth; Villus: Crypt: Villus Relationship: Crypt; GC: Globet cells.

The negative effects of high ambient temperature are known and have been reported by Marchini et al. [25], where high ambient temperature up to the fourth week of age promoted reduction in digestive enzyme secretion and in the VH of broilers.

According to some authors, food digestibility and intestinal mucosal integrity are strongly related to ambient temperature variations. Thus, the low amount of food present in the gastro-intestinal tract during thermal stress also impairs the trophic stimulation of the intestinal mucosa, besides decreasing the secretion of digestive enzymes [25, 26]. At high temperatures, feed intake is decreased in an attempt to decrease endogenous heat production that could cause damage to intestinal morphology and integrity, compromising digestion and absorption

**Table 4. Average productive performance and eggshell thickness of laying quail submitted to methionine supplementation at 3 levels (100, 110 and 120%) and 4 temperature ranges varying from thermoneutrality to heat stress.**

| Level x Temp | Production % 2° Period | | | |
|---|---|---|---|---|
| | 20°C | 24°C | 28°C | 32°C |
| 100% | 89.55±3.70aC | 92.2±4.42aB | 85.24±4.41aA | 83.81±4.94bA |
| 110% | 91.67±3.40aB | 91.27±6.02aB | 85.32±5.21abA | 76.98±11.35bB |
| 120% | 93.52±8.54abA | 98.23±9.78aA | 85.3±6.77bcA | 74.78±6.78cB |
| | Eggshell thickness 2° Period | | | |
| | 20°C | 24°C | 28°C | 32°C |
| 100% | 0.27±0.02aA | 0.25±0.01bB | 0.26±0.02abB | 0.25±0.01bB |
| 110% | 0.27±0.01aA | 0.25±0.01aB | 0.26±0.01aB | 0.26±0.02aA |
| 120% | 0.27±0.01aA | 0.26±0.01aA | 0.27±0.01aA | 0.26±0.01aA |

Means followed by the same lowercase letter in the rows and uppercase in the columns do not differ by Tukey's test up to 5% probability.

mechanisms and thereby reducing bird performance [26]. Thus, such a reduction may have led to lower methionine consumption at 32°C, and compromising the expected effect of methionine in reversing, at least in part, the deleterious effects of VH heat stress in both duodenum and jejunum. Animals kept at the lowest temperatures in this study maintained their feed intake, thus explaining the increase in duodenal and jejunum villus means. Therefore, for the villus height variable, methionine supplementation is recommended in cases of thermoregulation because it provides greater villus length, and consequently greater area of contact with nutrients and absorption, however, it would not be recommended in thermal stress situation.

Although methionine supplementation (120%) has not been shown to increase duodenal and jejunal villi under heat stress, and thus reverse the deleterious effects on intestinal morphology, it can be effective in increasing the absorption area (increased VH) in thermoneutrality.

Intestinal crypts are related to the intestinal health of the animal, the greater the crypt depth (CD), the greater the villous regeneration due to possible injuries (mechanical and/or other pathogenic mechanisms) occurred, or due to villous growth related to animal growth [27], becoming an important variable to be analyzed. Thus, the increase in CD can also predict an increase in VH when a trophic agent is presented, because it is exactly in this region that the cells that will migrate are produced to ensure the maintenance and/or increase of VH.

The histomorphometric analysis of CD in the duodenum showed that at thermoneutral temperatures for quail, the CD was lower, that is, the temperature of thermal stress (32°C) and at the lowest temperature (20°C), close to the thermal stress by low temperature, there is a higher need for cell turnover. However, 120% methionine supplementation led to a decrease in CD at higher temperatures (28 e 32°C). It can be inferred that at high temperatures, methionine supplementation reduced the deleterious effects of stress, reducing the need for cell proliferation in this region. The same result found for duodenum concerning methionine supplementation was found in the jejunum. Regarding temperature, the CD was different only at 20°C; it was lower at this temperature. The results show that methionine supplementation at high temperatures leads to a decrease in CD, which leads to a greater CVR, a variable used as an important marker of intestinal health, as it reveals a larger area of contact with food, consequently increased absorption without the need for too much energy expenditure on crypt turnover.

Crypt epithelium hyperplasia found at 32°C must have been induced to reestablish villus height, and is considered a compensatory mechanism [28], since thermal stress by heat in broilers for four consecutive days causes negative alterations in duodenum and jejunum

crypts, including reduction in villus height, in CVR, in absorption area, and increased crypt depth.

The VCR is related to the intestinal health of the animal, the higher the ratio, the greater its intestinal health. The results showed that the reduction of CD found in methionine supplementation at high temperatures (28 and 32˚C) in the duodenum and jejunum did not translate into higher VCR. In contrast, at 32˚C, VCR was lower in both intestinal segments after methionine supplementation. The improvement in intestinal health seen from the increase in VCR was only observed at a temperature of 24˚C at the duodenal level. These results show that methionine supplementation in laying quail under thermal stress does not reverse the deleterious effects of heat on VCR. However, these result presents similar to that found for villus height. In which we can infer that methionine supplementation is responsible for increasing the disponibility of nutrients, thus not needing to increase the area of contact with food, which would increase energy expenditure in an animal that is already ingesting less due to thermal stress.

These results corroborate Wu et al. [29], which report that thermal stress by heat is detrimental to the integrity of the intestinal mucosa of broilers, where the villi become shorter and flatter, and consequently the crypt increases its activity by becoming deeper in an attempt to reverse this situation, with this the VCR decreases. According to the authors, the high ambient temperature reduces feed intake of birds [29], justifying the reduction in VCR at high temperature. This leads to less energy available for maintenance and renewal of the intestinal mucosa and consequently for production.

Goblet cells (GC) play an important role in the digestive system of animals; the quantity of GC even indicates the degree of intestinal health. GC produces mucus, mucin, which protects the intestinal epithelium from infectious agents and mechanical agents, and forms the glycocalyx which also plays an important role in intestinal digestion [29]. It is well known, as happened in the present study at the duodenal level (reduction of GC at 32˚C), that high temperatures decrease the quantity and production of GC, thereby reducing intestinal health. Sandikciet al. [30] reported a significant reduction in GC in the three intestinal segments, in addition to villus height, in Japanese quail subjected to thermal stress. According to the authors, it is especially possible to relate the damage observed in the intestinal mucosa, including decreased GC, to low feed intake during thermal stress [31]. Unlike in the literature [30] for chickens, rising temperatures did not decrease jejunum GC, perhaps because quail are more heat tolerant than laying hens.

Methionine supplementation led to an increase in GC during thermal stress (32˚C) in the jejunum. These results demonstrate that methionine supplementation in heat stress reverses the deleterious effects of heat on the jejunum, but not on the duodenum. The increase in GC found in the jejunum under conditions that mimic thermal stress by heat provides better protection of the intestinal mucosa and better digestion, leading to improved intestinal health [32], thus justifying the use of methionine supplementation in this case.

Climate warming has been causing concern for quail farming, since, as results show, thermal stress by heat promotes various alterations in behavior and physiological mechanisms of quail, culminating in harmful morphological alterations, poor bird performance, and economic losses for the sector [33].

**Uterine histomorphometry.** For the first time in quail, morphometric results showed that the increase of temperature, that is, thermal stress (32˚C), causes decrease in the uterine folds, mainly in the secondary and tertiary folds (Table 3), which implies in a smaller area for the production of calcium carbonate, the main eggshell compound [2], negatively influencing the egg production of the animals. High temperatures also decreased eggshell production and thickness (Table 4). The highest indexes of uterine folds were found in treatments at 20 and 24˚C.

These results corroborate the egg production results (Table 4); the highest productive performance was found at 24˚C, and the lowest performance was at 32˚C.

Although the literature mentions that methionine supplementation increases the amount of folds in the uterus in layers and light birds [17–19], this was not observed when heat stress was applied in this study on quail. There was an increase in the uterine fold index only at 28˚C with 110% methionine supplementation; however, these results did not interfere with egg production (Table 4). We can infer that methionine supplementation, except 110% at 28˚C, does not minimize the deleterious effects of heat stress on uterine folds in quail.

**Measurement of hepatic glycogen and steatosis storage.** The increase in ambient temperature decreased the hepatic steatosis index, with the highest indexes at 20 and 24˚C. Methionine supplementation (120%) increased this rate only at 28˚C. This result differs partially from that found by Bunchasak and Silapasorn [16], who describe that the higher the methionine supplementation the higher the fatty acid synthesis in laying hens, and thus the higher the rate of hepatic steatosis. This variable is important since the increase in hepatic steatosis is related to estrogen production, that is, the higher the steatosis, the higher the estrogen production and the higher the egg production [16]. Thus, methionine supplementation (120%) at 28˚C would not only increase hepatic steatosis but also egg quality by these quail, which occurred in the present study, but not in production (Table 4). For this variable, methionine supplementation brings benefits at 28˚C, but not at 32˚C.

Hepatic glycogen levels did not appear to change with the alteration in ambient temperature, corroborating studies on broilers by Lana et al. [34]. However, this study was contrary to the results for broilers, which showed decreased hepatic glycogen stores, as well as reduced feed intake and weight gain [35], and reduced liver weight [36] with consequent reduction in metabolic activity under thermal stress. Probably no reduction in hepatic glycogen deposit was found in quail at 32˚C, as these animals have a greater tolerance to heat than broilers.

However, only at 24˚C, 110% methionine supplementation increased hepatic glycogen stores. These results demonstrate that at thermoneutrality, 110% supplementation maximizes energy storage in the form of glycogen in the liver. Such a surplus could be transferred to production, in this case in egg production, however it did not happened. Thus for this variable, in heat stress methionine supplementation was not efficient.

## Immunohistochemistry

In modern strains of laying hens, which can be extrapolated to laying quail, the equivalent of 10% of total body calcium is transferred daily for deposition as eggshells [37–39]; the major sources of calcium are through absorption from the diet at the intestinal level, renal resorption, and bone storage. Since Calbindin-D28k is the epitheial carrier responsible for the absorption of calcium from the digestive system, it would have the ability to modulate its deposition in the womb [40], in addition to intestinal absorptive capacity [41], influencing the production and the eggshell quality.

**Intestine.** For all treatments (temperatures and methionine levels), anti-Calbindin-D28k was positive throughout the duodenal epithelium; the lamina propria (connective tissue layer below the epithelium) was not positive (Fig 1). Positivity was more intense in the basal and more apical portion of the epithelium, since the middle portion was an area that had many enterocyte nuclei and the present marking is cytoplasmic. The most positive area was in the apical surface of the enterocytes. In contrast, goblet cells were not positive for anti-Calbindin-D28k. These results corroborate the study carried out in layers, which states that in the intestine of layers, there is Calbindin-D28k positivity (protein) in all segments, higher in the duodenum and jejunum, especially in the apical portions of the villi, but smaller in the ileum [39, 42].

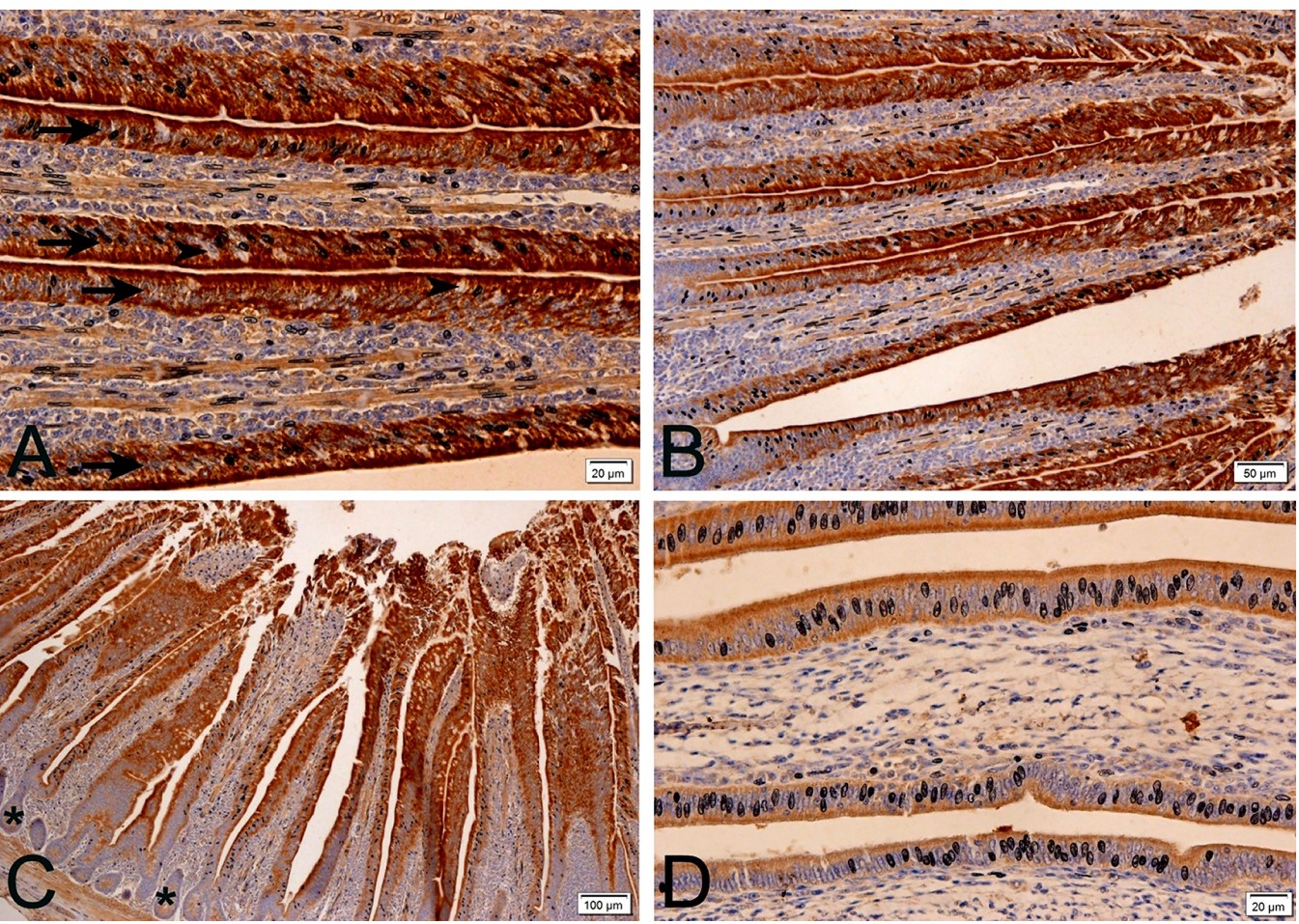

**Fig 1. Photomicrographs of anti-Calbindin-D28k immunohistochemistry in laying quail intestine at different magnifications.** Positive (brown stain) anti-calbindin-D28k intestinal epithelium (arrows) and non-positive goblet cells (arrowheads) (A and B) are observed. Non-antibody-positive crypts (asterisk) are also observed (C). Lower epithelial positivity (brown stain) is observed under 28˚C (D) when compared to other temperature treatments. Chromogen staining diaminobenzidine+hematoxylin.

Anti-Calbindin-D28k positivity in the duodenal intestine epithelium corroborates calcium absorption in this region. The greater intensity of positivity in the apical portion of enterocytes corroborates previous studies [39] that cite Calbindin-D28k as a cellular calcium transport. This transporter binds to calcium absorbed by the cell and diffuses it into the cytoplasm, which is finally extruded by $Ca_2 + $-ATPase into the basolateral membrane, reaching the vascular system through lamina propria vessels [39]. The form present in birds is 28kDa molecular weight, or Calbindin-D28k, present in the kidney, brain and intestine and uterus of birds [4, 7, 43, 44]. Goblet cells are not positive, since these cells have no function of absorption, but of production; they are responsible for producing and releasing mucin on the intestinal surface, as well as in other organs.

Among the temperatures studied, there was lower positivity to the epithelial calcium transport at 24 and 28˚C (Fig 1D), when compared to 20˚C, and mainly to 32˚C. At temperatures with lower positivity, the increase in methionine level had even lower positivity to anti-Calbindin-D28k. At 32˚C (heat stress), methionine supplementation (120%) also led to lower positivity for Calbindin-D28k when compared to the 100% level.

The decrease in positivity could be explained by the increased availability of calcium and consequently less need for absorption, and increased eggshell quality, which occurred in the present study. Although the performance model of Calbindin-D28k has already been described in layers, this is the first study in quail. In a study with methionine supplementation in diets with lower protein levels in Thailand, a country with thermal similarities to that of northeastern Brazil, there was an increase in laying production rates, including increased egg-shell thickness [16]. In the aforementioned study, the increase in methionine must also have minimized the deleterious effect of heat stress and increased calcium availability to improve production rates, as occurred in the present study. It can be imagined that in this study the positivity of Calbindin-D28k must also have decreased.

High temperature stress negatively affects laying performance, decreasing feed intake, live weight gain and efficiency [45, 46]. It also decreases egg production and eggshell quality and thickness [47–49] due to decreased availability of calcium ions [49]. It is important to say that increasing dietary calcium does not improve the quality of the shell in heat stress conditions [50, 51]. High temperature heat stress decreases the presence of Calbindin-D28k in the ileum, cecum, colon, and uterus of birds, causing deterioration of eggshell quality [52]. In the present study it was also observed the same effect on the duodenum; the heat stress decreases the positivity to Calbindin-D28k in the duodenum.

In the present study, at temperatures considered to be more thermoneutral for the quail (24 and 28˚C), the animals presented lower positivity of the cellular calcium transport in their duodenal epithelia, exactly because they were in better thermal comfort, and did not need a higher absorption of calcium (Fig 1). Literature provides studies [53] that show that the higher the ambient temperature and thermal stress, the greater the need to supplement dietary calcium, as the animals will need more calcium for their metabolism, since the thermal stress decreases the availability of calcium [49]. Within the 24 and 28˚C treatments, methionine supplementation provided even lower positivity for anti-Calbindin-D28k when compared to normal levels. The gene expression of this gene under methionine supplementation followed a similar result (Fig 5). Therefore, it is assumed that methionine supplementation leaves more calcium available, which makes the need for lower intestinal calcium absorption necessary.

The greatest positivity was at the temperature of thermal stress (32˚C), when it is thought to have less calcium available, which increases the need for calcium. Thus there were more calcium transporters (Calbindin-k28D) (higher positivity) to provide greater absorption to maintain the production. The positivity at the temperature of 20˚C is intermediate, because for quail, this temperature is already relatively low, thus, the animal already feels some result of thermal stress, in this case for low temperature, changing its physiology, and also needing more calcium. This explains the slightly higher positivity at 20˚C than that found in the 24 and 28˚C treatments.

This is the first study to describe Calbindin-D28k protein expression in the intestine of laying quail, and it is also the first to demonstrate the influence of high temperature heat stress on this epithelial calcium transport. The results related to the protein expression of Calbindin in the intestine suggests that methionine supplementation provides more calcium available for the animal, with no need for more carriers, physiologically supporting the indication of methionine supplementation for quail, even in situations of thermal stress.

**Kidney.** Calbindin exists in 2 major forms: with low molecular weight, a 9kDa protein (Calbindin-D9k) present in mammalian intestines, and another with the high molecular weight with 28kDa (Calbindin-D28k), present in the kidney of birds [7, 4] and mammalian kidneys [7]. In the present study, anti-Calbindin-D28k positivity was found in the distal contorted tubules (**DCT**) of the nephrons, but there is practically no positivity in the proximal contoured tubules (PCT). This positivity in DCT was more intense in the region surrounding

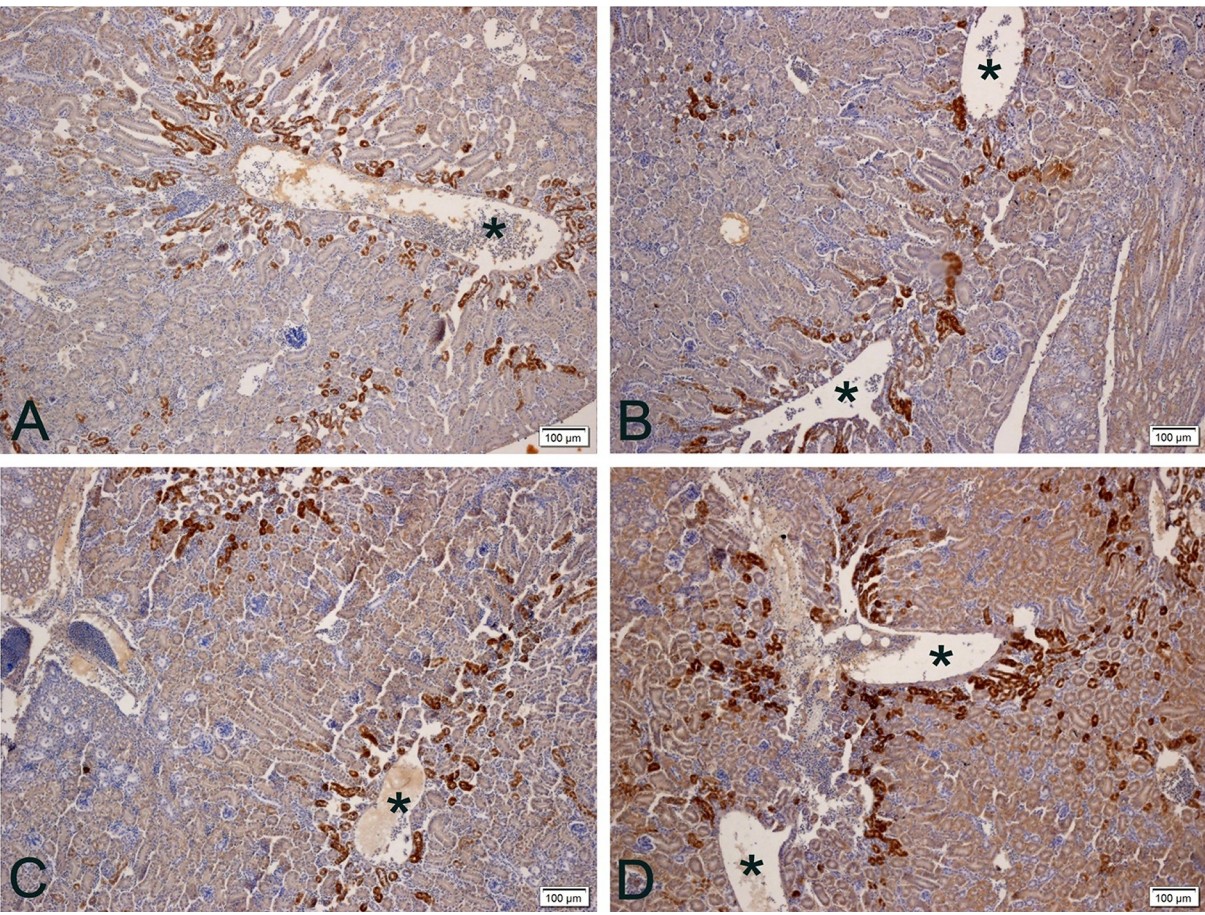

**Fig 2. Photomicrographs of anti-calbindin-D28k immunohistochemistry on the kidney of quail laying at different temperatures.** A) 20˚C: Positivity (brown stain) occurs mainly in distal contorted tubules near large vessels (asterisks). B and C) 24 and 28˚C: Lower antibody positivity (brown stain). D) 32˚C: There are more positive distal contorted tubules and slightly positive proximal contorted tubules as well. More positivity (brown stain) is observed at temperatures 20 and 32˚C. Chromogen staining diaminobenzidine + hematoxylin. Magnification 100x.

the large renal blood vessels in the renal cortex (Fig 2). This feature can be explained by the fact that these areas have blood with a higher amount of calcium, which has not yet been reabsorbed. The renal corpuscle, as well as the glomerulus (capillaries), was not positive for anti-Calbindin-D28k.

Calbindin-d28k positivity was higher in DCTs, as these are the sites of greatest mineral resorption, including calcium [54]. The PCT showed little positivity by not resorbing this mineral normally and in the renal corpuscle there is no positivity, as this portion of the nephron does not absorb or reabsorb, only filters blood resulting in pre-urine.

The positivity is lower at 24 and 28˚C compared to at 20 and 32˚C, just as it did in the intestine. In the treatment at 32˚C, the amount of positive DCT increased, always in greater numbers near the great renal veins (Fig 2). At 28 and 32˚C, PCT were more positive when compared to previous temperatures.

In the treatments in which the animals are theoretically in greater thermal comfort, that is, in the 24 and 28˚C treatments, the anti-Calbindin-D28k positivity was lower. Since stress by high temperatures negatively affects the performance of layers [45, 46] due to decreased availability of calcium ions [49], it is expected that the rate of renal resorption will have to increase under stress. Thus, animals in thermal comfort would have less need to reabsorb large

amounts of calcium, as occurred in treatments with 24 and 28°C (lower positivity). In animals with some degree of thermal stress, such as at 32°C (high temperature) and 20°C, theoretically because it is a temperature below the thermal conformation for the species, they would have greater need to reabsorb more calcium (higher positivity).

Although it is well known that thermal stress by high temperatures decreases the presence of Calbindin-D28k in the ileum, cecum, colon and uterus of birds, causing deterioration of eggshell quality [52], there was no information in literature on the influence of this calcium transport at renal level for thermal stress by high or low temperatures, as seems to occur at a temperature of 20°C. Thus, this is the first report on the influence of heat stress on such a mode of transport in the kidney, which, like the intestine, has the opposite effect to that found for other organs in other experiments with layers [52], but similar to that found in the present study in the intestine.

In the case of high temperature heat stress treatment, more DCT were positive for anti-Calbindin-D28k, which shows that under such a situation not even increased positivity was enough to reabsorb the calcium needed for the production of these birds; in addition to the increase in the expression of this transport, the increase in the number of DCT that expressed such transport was needed (Fig 3).

The most intriguing was that at the two higher temperatures (28 and 32°C), the positivity of PCT also increased, so it seems that the animal physiologically adapts to reabsorb calcium, not only by DCT but also by PCT in case of high temperature stress.

Methionine supplementation does not alter protein expression of the Calbindin-D28k. Such an increase may not be necessary, given the increased availability of calcium that methionine supplementation has provided.

**Uterus.** Positivity to anti-Calbindin-D28k was high in the uterine glands, since these are the sites of calcium carbonate production and secretion, which is produced and released for eggshell production in the uterus, and is influenced by increased circulating estrogen [40], and modulates eggshell production and quality. Uterine gland cells transport calcium from their basal portion to the apical surface during calcium carbonate production, the more calcium carbonate, the faster the egg production and/or better eggshell quality. The epithelium (ciliated pseudostratified) is not positive for anti-Calbindin-D28k except for a thin layer on the apical portion of this epithelium.

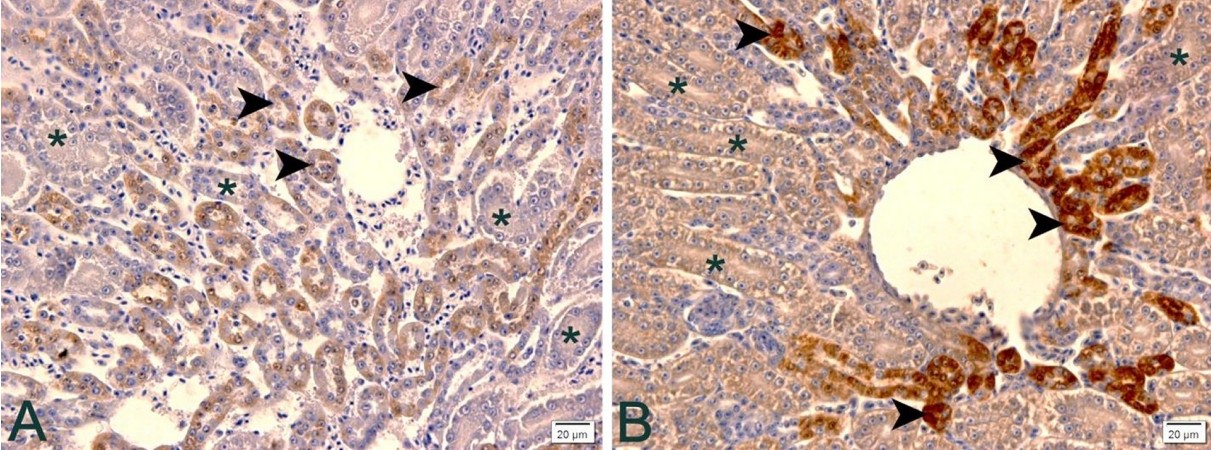

**Fig 3. Immunohistochemistry photomicrographs of anti-Calbindin-D28k in the kidney of laying quail at different temperatures.** A) 24°C: Note poor positivity (brown stain) in distal contorted tubules (arrowheads) and no positivity in proximal contorted tubules (asterisks). B) 32°C: In thermal stress there is intense positivity (brown stain) of the distorted contorted tubules (arrowheads) and low positivity (brown stain) in the proximal contorted tubules (asterisks). Chromogen staining diaminobenzidine + hematoxylin. Magnification 400x.

The positivity pattern was higher at 24 and 28°C, lower in the treatment in which the animals were submitted to 32°C, and intermediate at 20°C (Fig 4). Methionine supplementation by 120% increased anti-calbindin-D28k positivity.

Ebeid et al. [52] described that under conditions of thermal stress by high temperature there is a decrease in the presence of Calbindin-D28k in the uterus of laying hens, which corroborates the present study, in which the anti-calbindin-D28K positivity was lower at 32°C. This is the first time this fact is observed in quail. Stress by high temperatures is already known to negatively affect the performance of layers [45, 46] due to decreased availability of calcium ions [49]. This can be explained by heat stress reducing the conversion of vitamin D3 to its metabolically active form, 1.25 (OH) 2D3, which is essential for calcium absorption and utilization. Indeed, the calcium requirement for layers increases with high ambient temperatures [53]. This lower positivity to Calbindin-D28k observed provides less transport of calcium through the uterine glands and consequently less production of calcium carbonate, leading to a longer time for eggshell formation.

Methionine supplementation at high temperatures (32°C) promoted increased positivity of Calbindin-D28k in the uterine glands (Fig 5) reversing part of the deleterious effect of thermal stress. The increase in positivity of this transport under thermal stress conditions possibly increased uterine gland calcium carbonate excretion for eggshell production, improving egg quality, although not reaching thermal comfort values (Table 4). These findings also support physiologically the recommendation of methionine supplementation under conditions of thermal stress for quail.The fact that the positivity of the 20°C treatment was intermediate shows that perhaps for quail, which have high heat resistance, this temperature is already below the ideal temperature for them.

## PCR in real time (qPCR) for TRPV6 and Calbindin-D28k

TRPV6 acts as an epithelial channel of calcium in organs and glands that are characterized by high demand for calcium transport [5, 6, 54]. According to some studies [55, 56], this ion channel exerts a facilitator effect on calcium entry into epithelial cells, expressed in the intestinal and kidney absorption and resorption epithelia, but there is still little information about its expression pattern in laying hens [20], and none in laying quail. Calbindin, in turn, has been described in studies in its two main forms, Calbindin-D9k (low molecular weight protein) present in mammal intestines, and Calbindin-D28k (high molecular weight protein) in kidney, brain and intestine and uterus of birds [4, 7] and kidney of mammals [7].

Calbindin-D28k gene expression (Fig 5) in the kidneys of laying quail without methionine supplementation was the same for all temperatures, unlike the presence of the Calbindin-D28k protein, which was lower at 24 and 28°C. With methionine supplementation, the highest expression was at 24°C. By comparing the expression of this calcium carrier within each temperature (supplemented or not with methionine), it is possible to say that methionine supplementation only increases Calbindin-D28k gene expression at 24°C (thermoneutrality), that is, by supplementing methionine we can maximize calcium reabsorption at the renal level, which can increase egg production by producing thicker eggshells in less time. Unlike, the presence of the protein Calbindin-D28k in the tissue of the same organ, where no difference was observed with methionine supplementation. However, under conditions of heat stress, methionine supplementation did not increase this gene.

For TRPV6 gene in the same organ, the highest gene expression in animals without supplementation occurred at a temperature of 20°C, already mentioned as a temperature below thermal comfort for laying quail. In animals submitted to methionine supplementation, the highest peak of gene expression occurred at a temperature of 24°C (thermoneutral), coinciding

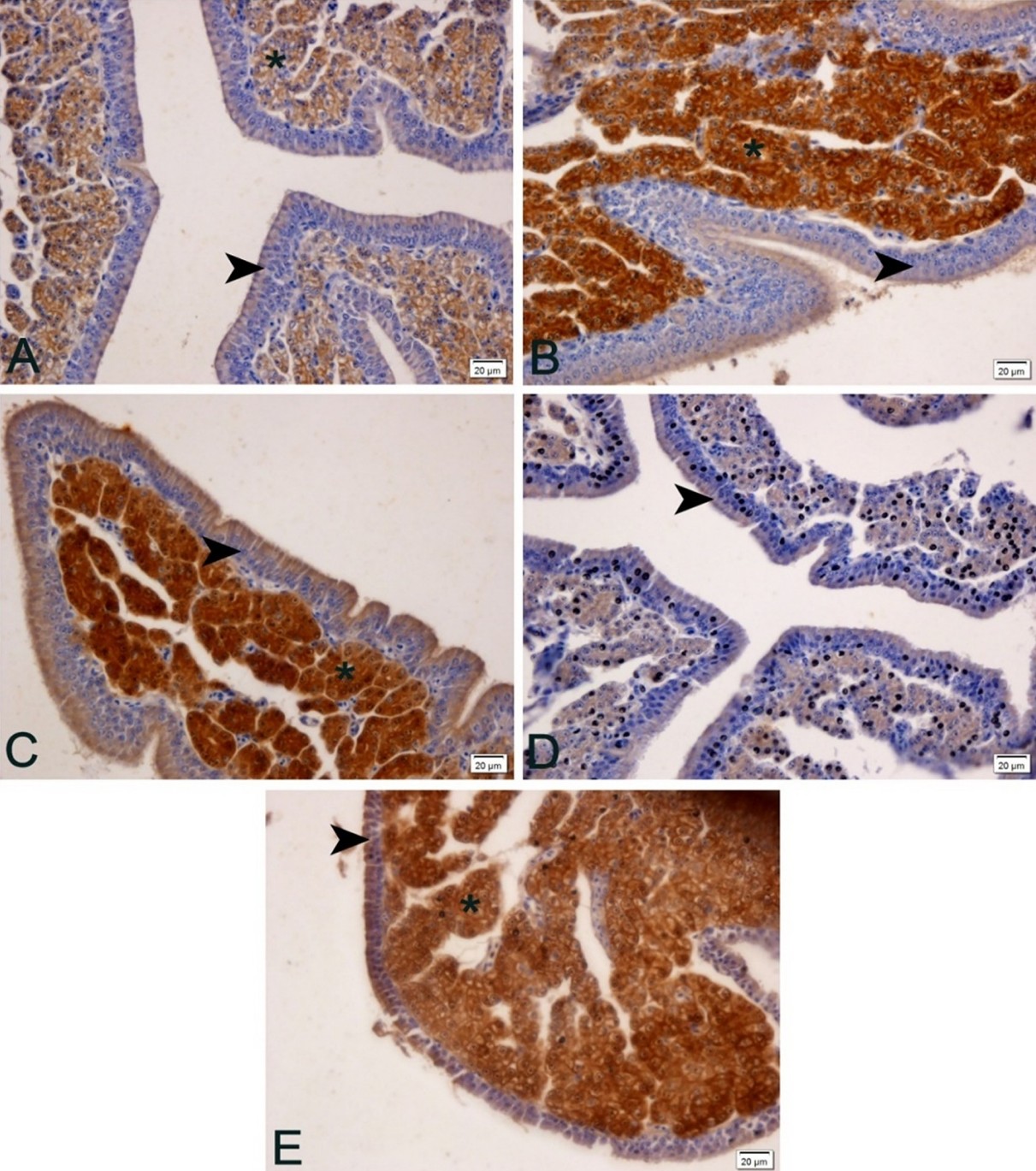

**Fig 4. Photomicrographs of anti-Calbindin-D28k immunohistochemistry in the uterus of laying quail at different temperatures (20, 24, 28 and 32˚C) and supplemented with 120% methionine at 32˚C.** A) 20˚C: observe lower positivity (brown stain) in the uterine glands (asterisk). B) 24˚C: observe greater positivity (brown stain) in the uterine glands (asterisk). C) 28˚C: observe greater positivity (brown stain) in the uterine glands (asterisk). D) 32˚C: observe lower positivity (brown stain) in the uterine glands (asterisk). E) 32˚C: supplemented with 120% methionine. More positive than treatment without methionine supplementation. Arrowheads (uterine epithelium), asterisks (uterine glands). Chromogen staning diaminobenzidine + hematoxylin staining. 400x magnification.

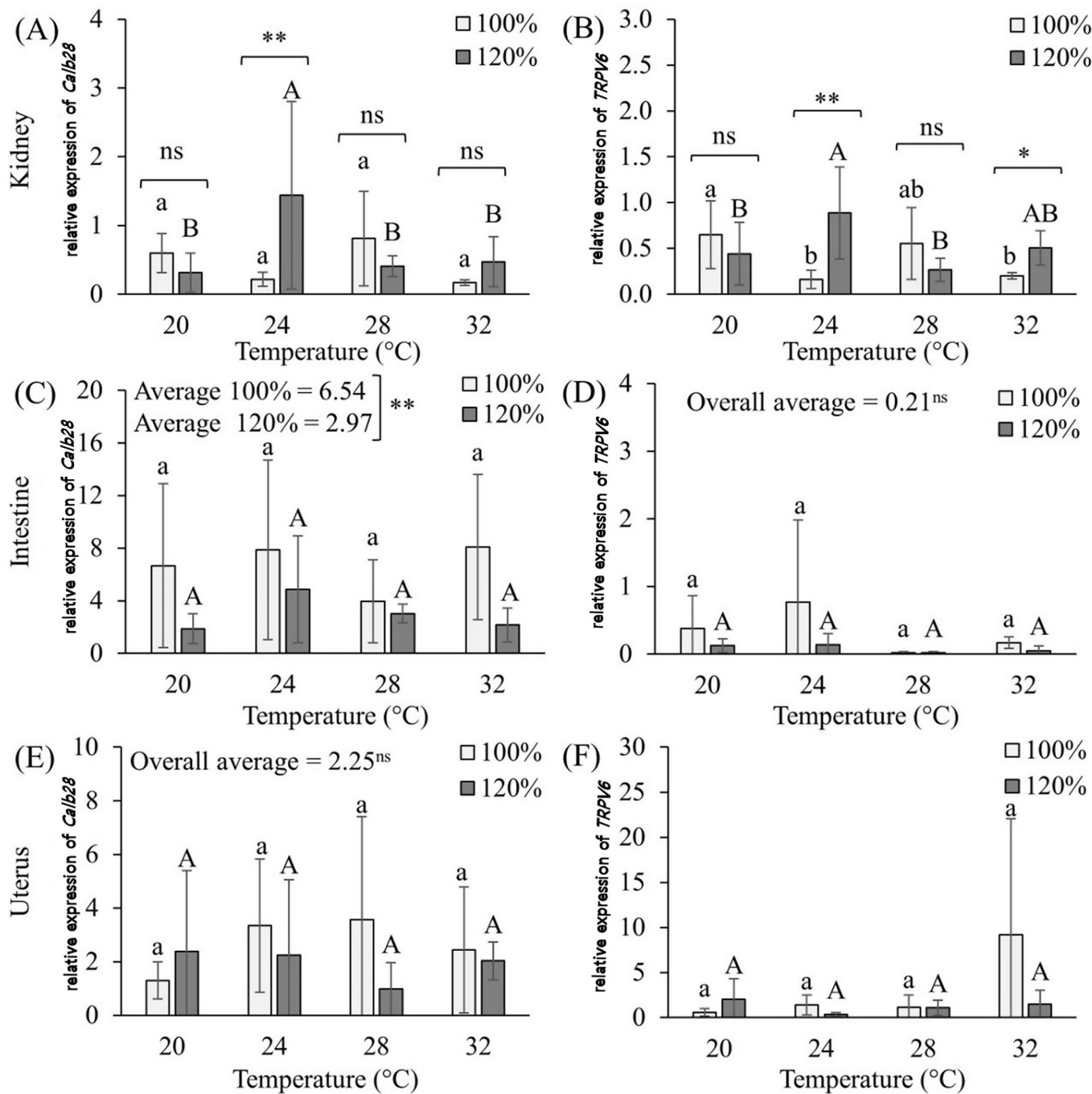

**Fig 5.** Graphs of the effects of methionine supplementation (100% and 120%) at different temperatures on the Calb 28 (A) and TRPV6 (B) genes expressions in the kidneys; Calb 28 (C) and TRPV6 (D) in the intestine; and Calb 28 (E) and TRPV6 (F) in the uterus of Japanese quail (*Coturnix japonica*) in production phase. [a,b,A,B] Averages followed by the same lower case letter for 100% supplementation and upper case for 120% supplementation do not differ from each other ($p<0,05$); [ns] not significant, [*]$p<0.05$, [**]$p<001$.

with the result obtained in the calbindin gene, followed by a temperature of 32˚C. Unlike calbindin, when comparing within each temperature (supplemented and not), we see that methionine supplementation increased TRPV6 gene expression, not only at 24˚C but also at 32˚C, that is, under thermal stress conditions. This result demonstrates physiologically and

**Table 5. Analysis of variance summary (mean squared) for the effects of different temperatures and methionine supplementation (100% and 120%) on *Calb* 28 and TRPV6 gene expressions in the kidneys, intestine and uterus of Japanese quail (*Coturnix japonica*) in the production stage.**

| Variation source | Gl | Kidney | | Intestine | | Uterus | |
|---|---|---|---|---|---|---|---|
| | | *Calb 28* | *TRPV6* | *Calb 28* | *TRPV6* | *Calb 28* | *TRPV6* |
| Temperature | 3 | 0.63ns | 0.12ns | 22.85ns | 0.54ns | 2.29ns | 61.66* |
| Supplementation | 1 | 0.62ns | 0.26ns | 197.63** | 0.93ns | 8.36ns | 45.33ns |
| Temp. vs Suplem. | 3 | 2.00** | 0.79** | 16.57ns | 0.29ns | 9.22ns | 54.33ns |

*p<0.05

**p<0.01, and ns not significant.

technically validates the use of methionine supplementation for laying quail in cases of thermal stress, and its effectiveness in minimizing the deleterious effects of high temperatures. Such an increase in this gene increases calcium reabsorption, making more of the mineral available for egg production, specifically in the release by the uterus for eggshell production.

These results corroborate studies that already cited the gene expression of both genes (TRPV6 and Calbidin-D28k) in kidney tissue of the laying birds [4, 7] and still stands out for being the first to demonstrate the positive expression of their gene expression, in the kidneys of laying quail.

In the intestine there was gene expression for both genes in all treatments, corroborating another study regarding the presence of Calbindin-D28k in layers [57]. This is also the first study to demonstrate TRPV6 gene expression in intestines of laying quail. However, it was not possible to observe gene alteration of Calbindin-D28k or TRPV6 with increasing temperature. However, methionine supplementation decreased the gene expression of Calbindin-D28k at the duodenum. These results imply that there is no alteration in the intestinal absorption or cellular calcium transport during thermal stress, but also it is assumed that methionine supplementation leaves more calcium available, which makes the need for lower intestinal calcium absorption necessary, as noted and already mentioned for Calbindin-D28k positivity through immunohistochemistry.

The TRPV6 gene, when compared to Calbindin-D28k, showed little expressiveness in intestinal tissue in all the treatments, however, contradicting and filling in the gap left by some authors [4, 57, 58], who claim that the presence of TRPV6 is still uncertain in birds, including layers or quail. Although the difference was not significant due to the high standard deviation, high temperatures seemed to decrease intestinal TRPV6 gene expression.

For uterine tissue, Calbindin-D28k and TRPV6 gene expression also occurred in all treatments, and as for intestine, it was poor for TRPV6. Heat stress and methionine supplementation did not influence gene expression of Calbindin-D28k or TRPV6. Differently of Calbindin-D28k positivity (protein) that decreased in thermal stress, and that increased with methionine supplementation. Since there was no change in gene expression, methionine supplementation would not be justified, at least when analyzing TRPV6.

The results described and observed in this study show the gene expression of TRPV6 and Calbindin-D28k genes in the renal, intestinal and uterine tissues of laying quail for the first time. Corroborating studies in laying hens [40–42] indicating that Calbindin-D28k would modulate the intestinal calcium absorption, as well as the significant presence of TRPV6 in the intestines and kidneys of layers [20].

Through analysis of variance, it was possible to verify the interaction between temperature and supplementation for both genes in the kidney (p≤0.01), supplementation for Calbindin-D28k in the intestine (p≤0.01) and temperature for TRPV6 in the uterus (Table 5).

The results obtained, justifies the use of 120% methionine supplementation in thermal stress in order to partially reverse the deleterious effects in the duodenal crypts, in the number of goblet cells in the jejunum, in the uterine folds, decreasing positivity of Calbindin-D28k in intestines and kidney, increasing the positivity of Calbindin-D28k in the uterus and increasing TRPV6 gene expression in the kidney of laying quail; and justifies the use of 120% methionine supplementation in termoneutrality condition in order to increase intestinal villus absorption area and height, increase steatosis, decrease positivity of Calbindin-D28k in the intestine and kidney, increase positivity of Calbindin-D28k in uterus, and increase gene expression of Calbindin-D28k the TRPV6 in the kidney of laying quail; besides justifing the methionine supplementation (120%) in laying quails for providing thicker eggshells at heat stress and for improving production (%) in thermoneutrality.

## Conclusion

This study brings for the first time histomorphological and expression variations (mRNA and protein) of TRPV6 and Calbindin-D28k in organs involved with absorption, reabsorption and calcium deposition in quail; justifies the methionine supplementation (120%) in laying quails for providing thicker eggshells at heat stress and for improving production (%) in thermoneutrality; and justifies the methionine supplementation (120%) in order to partially reverse the deleterious effects of heat stress in the epithelial calcium carriers, and in the digestory and reproductive morphology of laying quail.

## Supporting information

**S1 Data.**
(XLS)

**S2 Data.**
(XLS)

**S3 Data.**
(XLS)

**S4 Data.**
(XLS)

**S5 Data.**
(XLS)

**S6 Data.**
(XLS)

## Author Contributions

**Conceptualization:** Edilson Paes Saraiva, Fernando Perazzo da Costa, Ricardo Romão Guerra.

**Formal analysis:** Danila Barreiro Campos, Ricardo Romão Guerra.

**Investigation:** Lanuza Ribeiro de Moraes, Maria Eduarda Araújo Delicato, André da Silva Cruz, Hugo Thyares Fonseca Nascimento Pereira da Silva, Clara Virgínia Batista de Vasconcelos Alves, Danila Barreiro Campos, Edilson Paes Saraiva, Fernando Perazzo da Costa, Ricardo Romão Guerra.

**Methodology:** Lanuza Ribeiro de Moraes, Maria Eduarda Araújo Delicato, Hugo Thyares Fonseca Nascimento Pereira da Silva, Clara Virgínia Batista de Vasconcelos Alves, Danila Barreiro Campos, Edilson Paes Saraiva, Fernando Perazzo da Costa, Ricardo Romão Guerra.

**Project administration:** Ricardo Romão Guerra.

**Supervision:** Fernando Perazzo da Costa, Ricardo Romão Guerra.

**Validation:** André da Silva Cruz, Danila Barreiro Campos, Edilson Paes Saraiva, Ricardo Romão Guerra.

**Writing – original draft:** Ricardo Romão Guerra.

**Writing – review & editing:** Ricardo Romão Guerra.

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
