## [Decision Letter · Decision Letter 0]

24 Sep 2020

PONE-D-20-19659

Methionine supplementing effects on intestine, liver and uterus morphology, and on positivity and expression of calbindin-D28k and TRPV6 calcium carriers in laying quail in thermoneutral conditions and under thermal stress

PLOS ONE

Dear Dr. Guerra,

Thank you for submitting your manuscript to PLOS ONE. After careful consideration, we feel that it has merit but does not fully meet PLOS ONE’s publication criteria as it currently stands. Therefore, we invite you to submit a revised version of the manuscript that addresses the points raised during the review process.

While 2 reviewers were quite positive and have no further comments, 1 reviewer has raised issues which mainly relate to the presentation of the data.  Also, on reading the manuscript editor felt that the English language needs attention.  Minor issues e.g. Calbindin is being termed as chloride channel in abstract and at other places, which is not entirely correct.  Authors need to carefully correct the language as well as terms used which are not appropriate.  At this point no additional experiments are needed.

We look forward to receiving your revised manuscript.

Kind regards,

Pradeep Dudeja

Academic Editor

PLOS ONE

Journal Requirements:

2. To comply with PLOS ONE submissions requirements, in your Methods section, please provide additional information on the animal research and ensure you have included details on (1) methods of sacrifice, (2) methods of anesthesia and/or analgesia, and (3) efforts to alleviate suffering.

3.Thank you for stating the following financial disclosure:

 [No].

4.Thank you for stating the following in your Competing Interests section: 

[No].

Reviewers' comments:

Reviewer's Responses to Questions

**Comments to the Author**

1. Is the manuscript technically sound, and do the data support the conclusions?

Reviewer #1: Yes

Reviewer #2: Yes

Reviewer #3: Partly

2. Has the statistical analysis been performed appropriately and rigorously? 

Reviewer #1: Yes

Reviewer #2: Yes

Reviewer #3: Yes

3. Have the authors made all data underlying the findings in their manuscript fully available?

Reviewer #1: Yes

Reviewer #2: Yes

Reviewer #3: Yes

4. Is the manuscript presented in an intelligible fashion and written in standard English?

Reviewer #1: Yes

Reviewer #2: Yes

Reviewer #3: Yes

5. Review Comments to the Author

Reviewer #1: The authors address all of my concerns adequately. I still have a concern with the use of a single housekeeping gene. Did the authors establish steady state expression of actin in all tissues, organs at all treatments?

Reviewer #2: The authors presented data showing the effects of methionine enriched diet on intestinal epithelia and other tissues in cases of normal or high ambient temperature. The studies are sound and the conclusion is supported with convincing data. The authors also adequately addressed the conners of the previous reviewers.

Reviewer #3: Authors address an interesting scientific question regarding methionine supplementation for heat stress. The data is clear as discrete parts, but it is hard to understand how all the pieces fall under the authors overall conclusion that methionine supplementation is beneficial (even with the further specification that it is only beneficial for egg laying). Because of the lack of a clear train of thought, minor revisions to organize the paper into a more cohesive story is requested. No new studies are required.

1) A strong piece of data in this paper is the correlation of villus height to heat stress and in response to methionine. In high heat (32C), the quails had shorter villi in their intestine and this was further shortened by methionine supplementation. Please add to the discussion or conclusion how this fits in with the other phenotypes.

2) It is unclear how the staining of calbindin (fig 1-4) contributes to the authors' conclusions. The staining is quite dark with no definitive staining controls. Additionally, the accompanying text describes the staining pattern but does not define nor quantify the staining in terms of heat stress.

3) QPCR data on both targets does show trends that 20% extra methionine can affect expression. Perhaps instead of organizing the figures based on technique, the authors may think about each organ separately and add the IHC with corresponding QPCR data.

4) Figure 6 does not justify the methionine supplementation as written in the text. It appears to demonstrate that the calbindin and trpv6 are only related in kidney expression. It is recommended to omit this figure.

6. PLOS authors have the option to publish the peer review history of their article (what does this mean?). If published, this will include your full peer review and any attached files.

Reviewer #1: No

Reviewer #2: No

Reviewer #3: No

---

## [Decision Letter · Decision Letter 1]

5 Jan 2021

Methionine supplementing effects on intestine, liver and uterus morphology, and on positivity and expression of Calbindin-D28k and TRPV6 epithelial calcium carriers in laying quail in thermoneutral conditions and under thermal stress

PONE-D-20-19659R1

Dear Dr. Guerra,

We’re pleased to inform you that your manuscript has been judged scientifically suitable for publication and will be formally accepted for publication once it meets all outstanding technical requirements.

Kind regards,

Pradeep Dudeja

Academic Editor

PLOS ONE

Additional Editor Comments (optional):

Reviewers' comments:

Reviewer's Responses to Questions

**Comments to the Author**

1. If the authors have adequately addressed your comments raised in a previous round of review and you feel that this manuscript is now acceptable for publication, you may indicate that here to bypass the “Comments to the Author” section, enter your conflict of interest statement in the “Confidential to Editor” section, and submit your "Accept" recommendation.

Reviewer #2: All comments have been addressed

Reviewer #3: All comments have been addressed

2. Is the manuscript technically sound, and do the data support the conclusions?

Reviewer #2: Yes

Reviewer #3: Yes

3. Has the statistical analysis been performed appropriately and rigorously? 

Reviewer #2: Yes

Reviewer #3: Yes

4. Have the authors made all data underlying the findings in their manuscript fully available?

Reviewer #2: Yes

Reviewer #3: Yes

5. Is the manuscript presented in an intelligible fashion and written in standard English?

Reviewer #2: Yes

Reviewer #3: Yes

6. Review Comments to the Author

Reviewer #2: The concerns were adequately addressed. The authors were careful in responding to the reviewers. I have no further comments.

Reviewer #3: Thank you for addressing the comments and making changes to the manuscript. While the manuscript is quite jumbled without a clear organization of the data, the data itself is important and needs to be shared with the scientific community.

7. PLOS authors have the option to publish the peer review history of their article (what does this mean?). If published, this will include your full peer review and any attached files.

Reviewer #2: No

Reviewer #3: No

---

## [Editor Report · Acceptance letter]

7 Jan 2021

PONE-D-20-19659R1 

Methionine supplementing effects on intestine, liver and uterus morphology, and on positivity and expression of Calbindin-D28k and TRPV6 epithelial calcium carriers in laying quail in thermoneutral conditions and under thermal stress 

Dear Dr. Guerra:

I'm pleased to inform you that your manuscript has been deemed suitable for publication in PLOS ONE. Congratulations! Your manuscript is now with our production department. 

Kind regards, 

on behalf of

Dr. Pradeep Dudeja 

Academic Editor

PLOS ONE